# Superhydrophilic Modification of Polycarbonate Substrate Surface by Organic Plasma Polymerization Film

**DOI:** 10.3390/ma15134411

**Published:** 2022-06-22

**Authors:** Kuan-Wei Lu, Yu-Tian Lin, Hung-Sen Wei, Chien-Cheng Kuo

**Affiliations:** Department of Optics and Photonics, Thin Film Technology Center, National Central University, 300, Chung Da Rd., Chung Li, Taoyuan 32001, Taiwan; s1011159s@gmail.com (K.-W.L.); walter87501@gmail.com (Y.-T.L.); hswei@dop.ncu.edu.tw (H.-S.W.)

**Keywords:** superhydrophilicity, plasma polymerization, surface modification, polycarbonate, amino silane

## Abstract

Superhydrophilicity performs well in anti-fog and self-cleaning applications. In this study, polycarbonate substrate was used as the modification object because of the low surface energy characteristics of plastics. Procedures that employ plasma bombardment, such as etching and high surface free energy coating, are applied to improve the hydrophilicity. An organic amino silane that contains terminal amine group is introduced as the monomer to perform plasma polymerization to ensure that hydrophilic radicals can be efficiently deposited on substrates. Different levels of hydrophilicity can be reached by modulating the parameters of plasma bombardment and polymerization, such as plasma current, voltage of the ion source, and bombardment time. The surface of a substrate that is subjected to plasma bombarding at 150 V, 4 A for 5 min remained superhydrophilic for 17 days. After 40 min of Ar/O_2_ plasma bombardment, which resulted in a substrate surface roughness of 51.6 nm, the plasma polymerization of organic amino silane was performed by tuning the anode voltage and operating time of the ion source, and a water contact angle < 10° and durability up to 34 days can be obtained.

## 1. Introduction

Hydrophilic/hydrophobic materials have attracted extensive attention due to their unique properties and practical applications in the energy and environmental fields, and they have been successfully applied to material surface modification and film formation processes [1], with applications ranging from adhesion [2], self-cleaning and anti-fogging [3,4], optical components [5,6], biomedical materials, and compatibility [7,8]. Superhydrophilicity is defined as a water contact angle of less than 10° on the surface at a triple point. Therefore, a superhydrophilic surface helps achieve rapid drying and defogging.

Common hydrophilic materials, such as photo-induced photocatalyst [9] and compounds with high surface energy functional groups, are often used for such modifications. However, photocatalyst materials such as titanium dioxide (TiO_2_) require illumination, and their intrinsic refractive index and physical thickness must be considered in some optical applications. 3-Aminopropyltriethoxysilane (APTES), an alkoxysilane with high surface energy terminal amines [10,11], is frequently used as a coupling agent for attaching organic molecules to hydroxylated substrates, adhesion promotion, and biological implants [12]. Silanization allows APTES to attach to various surfaces [13]. This process only modifies the surface of the substrate, allows APTES to attach to the surface, and provides terminal groups with high surface energy, thereby being suitable as a precursor for improving the hydrophilicity.

Plastic substrates have been a trend for optical components in the past few years, and their advantages include a low cost, flexibility, and availability for large area processes. However, as a result of their low surface free energy, plastic substrates are hydrophobic or unattachable to optical thin films. The aging [14] phenomenon of polymer substrates is also susceptible to environmental conditions or plasma bombardment.

The material surface can be made superhydrophilic in various ways, such as ultrasonic spray pyrolysis, hydrothermal treatment, the sol–gel method, electrochemical anodization, surface etching, and spin coating [15]. However, such wet processes often produce environmental pollutants as a result of reactions in open spaces. The plasma polymerization process can provide highly functionalized organic surfaces. Many recent plasma polymerization studies have been applied to micro and nanotechnology, such as various kinds of surface functionalization or modification [16], biomedical science [17,18], and sensor applications [19,20].

Considering the direct environmental hazards of polluting by-products produced by the wet process and its complicated synthesis process, this research adopts ion source plasma bombardment and vapor deposition plasma polymerization. Reactive gas was introduced into a high vacuum chamber. After the substrates were etched and pretreated with ion source, the precursor was dissociated and polymerized to modify the surface of the polycarbonate (PC) substrate. Thus, surface uniformity, low cost, environmentally friendly, and a completely reacted procedure can be achieved.

## 2. Materials and Methods

APTES monomer (purity > 99%, Alfa Aesar, Haverhill, MA, USA) was applied as the precursor of the superhydrophilic thin film. Vaporized APTES can be pushed into the system by passing through nitrogen (N_2_) as the carrier gas. Optical-grade PC was used as the substrate. The PC sheet was first cut into a size of 30 mm × 30 mm × 2 mm, and the substrates were ultrasonically oscillated and cleaned by a mixture of deionized water and ethanol. Then, the residue was purged with nitrogen.

The ion source system (Vecco ion gun system with Mark II+ controller) illustrated in Figure 1 was used to dissociate reactive gases and simultaneously activate the substrate surface. The evacuated cavity pressure of the coating system is about 4 × 10^−4^ Pa. Adjusting the ion source current, voltage, and time enables argon (Ar) and oxygen (O_2_) to dissociate and chemically react with the surface of the PC substrate. Therefore, the surface of the substrate can be etched and activated by plasma to produce a large area of transient activation [21]. The flow rate of Ar and O_2_ is set as 2 sccm and 12.5 sccm, respectively, to achieve the most stable plasma condition. The working pressure is about 3 × 10^−2^ Pa when Ar/O_2_ is introduced into the chamber.

The surface free energy of the plasma-treated substrates increases, but the persistence is still poor. To promote the hydrophilicity and durability at the same time, superhydrophilic polymer film is necessary. Gaseous APTES (80 sccm) carried by N_2_ (5 sccm) was pumped into the high vacuum chamber and then deionized by the ion source to produce plasma, which will be deposited on the treated substrate. The working pressure when N_2_/APTES is introduced is about 2 × 10^−1^ Pa.

The following instruments were used to measure and characterize the hydrophilic properties, surface morphology, and composition. A water contact angle measuring instrument (FTA-125, First Ten Angstroms) was used to observe the hydrophilicity and evaluate the surface free energy of the samples. The morphology and surface roughness (Rq) of the substrates and films were characterized by using a Nanoscope atomic force microscope from Veeco Ins. Surface element composition ratio analysis was performed using X-ray photoelectron spectroscopy (XPS, Sigma Probe, Thermo-VG Scientific, Waltham, UK) with a Microfocus Monochromator Al anode X-ray. The plasma component was observed at different parameters by using an optical emission spectrometer (OES, AvaSpec-ULS2048, Avantes BV, Apeldoorn, The Netherlands) for plasma phase diagnostics.

## 3. Results

### 3.1. The Influence of Plasma Bombarding on Substrate Surface

Table 1a shows the effect of modulating different plasma currents on the surface roughness of the substrate with a fixed voltage of 250 V and a bombardment time of 20 min. The number of dissociated plasma particles increases as a result of the increased plasma current so that under the same area, the substrate surface is subjected to physical bombardment and chemical etching [22], and the substrate surface roughness increases. Bombarding the PC substrate at higher voltage plasma with the increasing current resulted in poor heat dissipation on the substrate surface and surface melting, thereby causing roughness reduction. Table 1b shows the effect on the surface roughness with a fixed voltage of 250 V and the best plasma current of 4 A in the previous section with different etching times. As a result of the increased bombardment time, the total number of particles per unit area subject to physical bombardment and chemical etching increased, resulting in a more obvious structure of the etched surface and improved roughness [23].

The ion source voltage will affect the gas dissociation rate, which is why the extent of cross-linking on the surface of the PC substrate after the dissociation of the monomers is different. Figure 2 shows the relationship between the influence of different voltages on the sample surface roughness with a fixed plasma current of 4 A and a bombardment time of 40 min. The roughness of the PC substrate gradually increases as the ion source voltage increases to 150 V, but it begins to decrease when the voltage exceeds 150 V.

According to Ting et al. [24], the cross-linking and chain scission of polymers occur simultaneously during the plasma bombarding process. Ion bombardment will cause cleavage and depolymerization of the polymer backbone and side chains [25,26]. Hydrogen ions are generated as a result of the cleaved chemical bonds in the polymerization, enabling the polymer to cross-link with other adjacent polymers. The degree of cross-linking depends on the chemical bonding of the polymer, the type of gas introduced into the plasma system, and the energy of ion bombardment. In addition, the process of ion bombardment can help the surface improve the bonding force before grafting the nitrogen-containing functional group. After the gases pass into the plasma system and are dissociated, the ultra-shortwave ultraviolet light radiates to destroy the original bonding on the substrate surface, thereby enabling free radicals to be grafted on its unoccupied chemical bonds [27,28,29].

In 1936, Wenzel proposed to modify the model of Young’s equation to explore the wetting behavior of liquid on rough surfaces [30]:(1)$$\cos\theta_{w}=\frac{A_{act}}{A_{pro}}\times \frac{\gamma_{SG}-\gamma_{SL}}{\gamma_{LG}}=r\cos\theta$$
where *r* (the roughness factor) is the ratio of the actual area ($A_{act}$) to the projected area $A_{pro}$) of a surface, and it is always greater than unity [31]. From the formula, we can conclude that a rough surface corresponds to a flat and large surface area of the water droplets. Therefore, the sample that has a larger rough surface can provide more exposed hydrophilic functional groups, thus resulting in the improved durability of hydrophilicity.

Figure 3 shows the hydrophilicity durability of plasma-treated PC substrates at different voltages. The ratio of gas flow rate is fixed, and the process is performed in 5 min, with the anode current controlled at 4 A. The longest durability can be achieved at an ion source voltage of 130 V. The increase in voltage can increase the number of days that the water contact angle persists. However, as time passes, the hydrophilic hydroxyl groups (−OH) generated by plasma bombardment on the surface gradually lose activity because of the aging phenomenon of the polymer substrate. Thus, the error of the water contact angle will increase.

### 3.2. Effect of Voltage on Organic Coating on Hydrophilicity and Durability

Given the poor durability of increasing surface energy after ion bombardment, the polymer film still needs to be deposited even though the PC substrates can be hydrophilic. Gueye et al. [32] used a microwave plasma system and introduced argon, nitrogen, and APTES monomers into the reaction to obtain terminal amine groups (−NH_2_) by selecting the best experimental conditions.

To prove the surface modification of this research, Figure 4 presents the emission intensity of each radicals from the dissociated monomers measured at different ion source voltages. The intensity of the free radicals from the plasma was measured by optical emission spectrometer; then, it was quantified and normalized by the emission intensity of Ar plasma. The content of free radicals that dissociated under different ion source voltages can be measured from the in situ emission spectrum of Ar/APTES plasma. At the ion source voltage of 130 V, the N-H radical signal intensity is the highest, indicating that the APTES cleaves and offers the greatest number of N-H radicals in the current plasma environment, resulting in the hydrophilic functional group having a higher probability of grafting on the surface bonding of the substrate. In addition, the signal intensity of C-N group and 2nd N^2+^ is the highest, indicating that more nitrogen ions in the plasma polymerization process can form CN radicals with the fragmented carbon atoms, have a higher probability of grafting on the substrate, and are likely to synthesize an amine or imine group with hydrogen ions. This condition contributes to an increase in durability.

XPS measurement was used to measure the polymer film, thereby doubly verifying the element ratio. The polymer film of APTES is superhydrophilic because the surface has exposed nitrogen-containing functional groups—amine (C-NH_2_), imine (C=NH), and sulfhydryl (C=N-OH)—to generate hydrogen bonds with water [33], with the nitrogen-containing functional group having the highest amine group content [34]. Therefore, with more amine groups being formed during the plasma polymerization process, the probability of grafting of chemical bonding on the surface increases, and the hydrophilic ability improves. Table 2 shows the surface elemental contents of the plasma polymer film (C1s, N1s, O1s, and Si2p) measured by XPS, indicating that the N/Si ratio is the highest at the ion source voltage of 130 V. The hydrophilic functional groups exposed on the polymer film are proven to be the greatest in number, thereby verifying that the choice of the ion source voltage is related to the nitrogen-containing functional groups and hydrophilic durability. This result matches that of hydrophilicity persistence (Figure 4).

### 3.3. Effect of Coating Time of Polymer Film on Durability

The experiment results of Lecoq et al. [34] showed that APTES polymer film carried by nitrogen without other gases has the greatest organic and amine content and the lowest deposition rate. The polymer film can be regarded as a modified layer by grafting functional group molecules on the PC surface. Figure 5 shows the durability of the hydrophilicity of polymer films with different coating times. A longer time results in improved hydrophilic durability. The film thickness is simply a layer of compounds, which is why oxygen in the environment easily influences the film, thus resulting in poor durability. Therefore, increasing the coating time to increase the density improves the durability of hydrophilicity.

### 3.4. Hydrophilicity of Surface-Modified PC Substrate

The polymer film can be deposited more effectively on the surface of the substrate, and the functional groups are exposed by using the parameters of 130 V, 4 A, and 25 min of APTES film coated after the surface is bombarded at 150 V, 4 A for 40 min. Figure 6 presents the hydrophilic durability of PC substrates treated by all processes, having a superhydrophilic persistence of up to 34 days. The modification process can be considered a relatively advanced technology for hydrophilic surface treatment for massive-area plastic substrate.

## 4. Conclusions

A superhydrophilic polymer film was prepared in this study by plasma polymerization on the surface of PC substrates. With the use of APTES as the reaction gas, a highly permanent superhydrophilic polymer film was obtained by controlling the parameters of surface modulation.

Ar/O_2_ was used to increase the substrate surface roughness and simultaneously activate the surface bonding via plasma bombardment. With the increase in plasma current and etching time, the surface roughness is effectively increased. The use of 150 V, 4 A, and 40 min of oxygen argon plasma in particular enables the PC substrate to achieve the maximum roughness (81.9 nm). However, at a voltage of more than 150 V, the surface of the substrate will exhibit an aging phenomenon because of the characteristics of its polymer, thereby decreasing its roughness.

In the preparation of the superhydrophilic polymer film, OES is used for real-time monitoring. Results show that more exposed nitrogen-containing hydrophilic functional groups deposited on the film, resulting in better hydrophilicity. When the ion source voltage of APTES is 130 V, the substrate surface has the highest ratio of CN and NH bonds. XPS analysis showed the highest proportion of nitrogen-containing hydrophilic functional groups on the surface (11.8%). The durability of the superhydrophilicity was the highest (34 days) after plasma treatment (150 V, 4 A, 40 min) and the deposition of APTES film (130 V, 4 A, 25 min).

## Figures and Tables

**Figure 1 materials-15-04411-f001:**
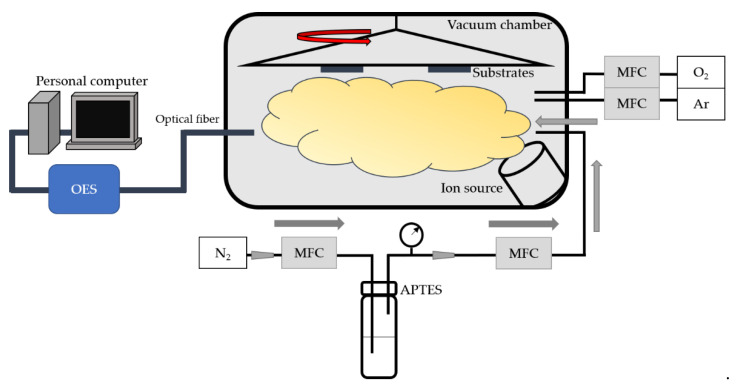
Ion source plasma bombarding and coating system.

**Figure 2 materials-15-04411-f002:**
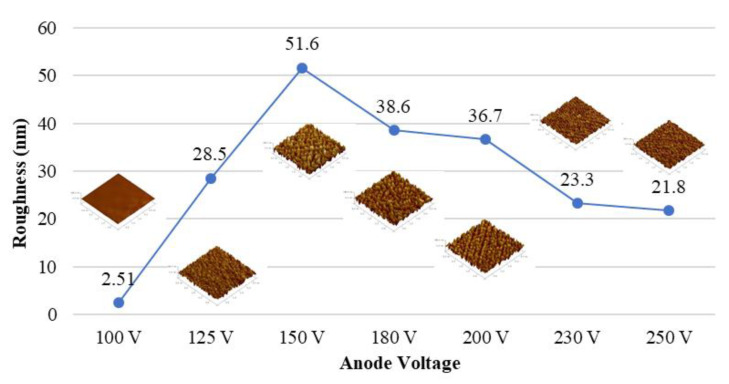
Roughness of PC substrates after different ion bombarding voltages.

**Figure 3 materials-15-04411-f003:**
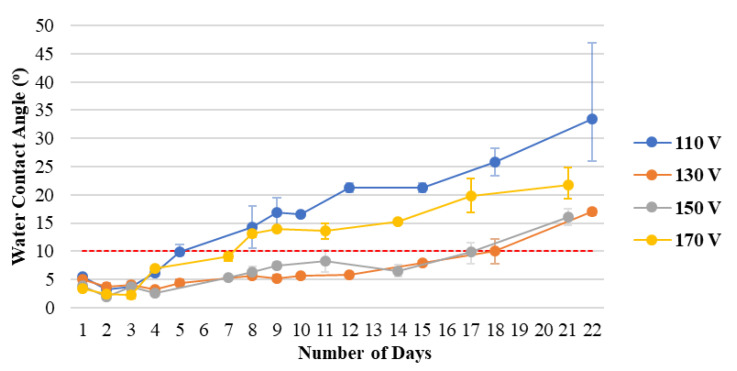
Hydrophilicity persistence of plasma-treated PC substrates at different voltages.

**Figure 4 materials-15-04411-f004:**
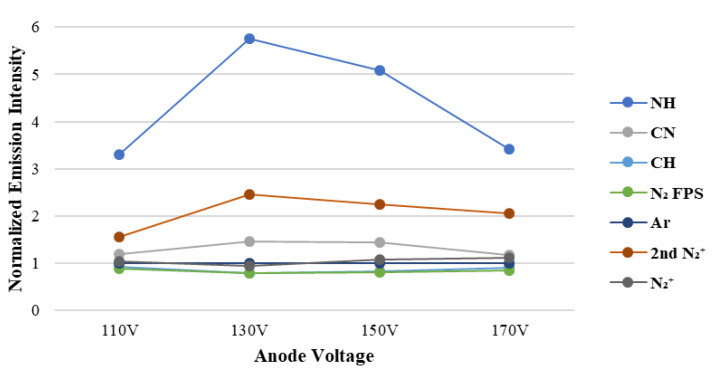
Radical contents of Ar/APTES plasma at different ion source voltages. (The emission intensity of each radical is normalized by the intensity of Ar plasma.)

**Figure 5 materials-15-04411-f005:**
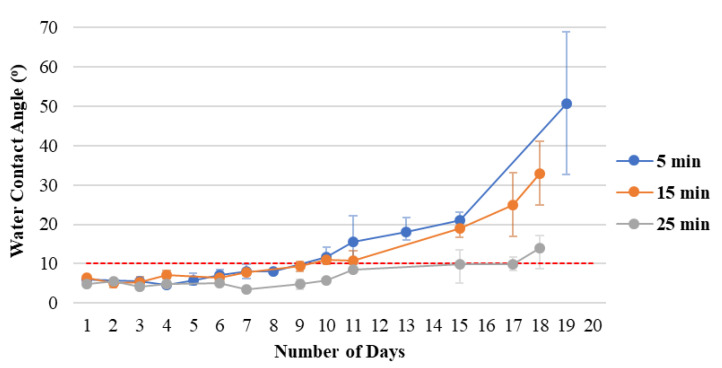
Hydrophilicity persistence of polymer films on untreated PC substrates with different coating times.

**Figure 6 materials-15-04411-f006:**
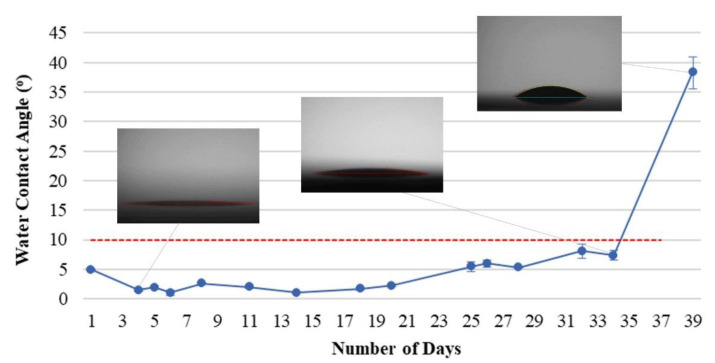
Hydrophilicity persistence of all-treatment PC.

**Table 1 materials-15-04411-t001:** Effect of (a) ion source currents, (b) bombardment time on surface roughness.

(a) Current	2 A	3 A	4 A
Surface Morphology	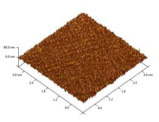	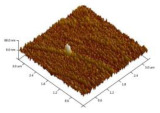	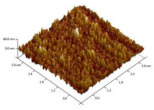
Roughness	5.13 nm	9.34 nm	13.3 nm
(b) Time	0 min	20 min	40 min
Surface Morphology	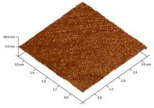	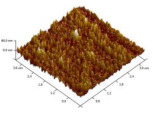	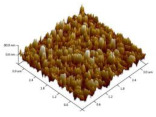
Roughness	2.09 nm	13.34 nm	22.1 nm

**Table 2 materials-15-04411-t002:** Element composition ratio of plasma polymer film on PC substrate.

Anode Voltage	C (at %)	N (at %)	O (at %)	Si (at %)	C/Si	N/Si	O/Si
110 V	57.28	6.96	27.87	7.89	7.26	0.88	3.53
130 V	55.15	9.08	30.67	5.10	10.81	1.78	6.01
150 V	54.35	8.77	30.83	6.05	8.98	1.45	5.10
170 V	53.93	9.82	29.07	7.17	7.52	1.37	4.05

## Data Availability

The data presented in this study are available on request from the corresponding author.

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
