# Peer review of "Superhydrophilic Modification of Polycarbonate Substrate Surface by Organic Plasma Polymerization Film"

_materials, 2022, doi:10.3390/ma15134411_

Round 1
Reviewer 1 Report
The paper from Lu et al. investigates the effects of plasma bombardment parameters on the hydrophilicity of polycarbonate. To this aim, different experimental parameters are considered; besides, an organic film deriving from APTES is applied on the polymer surface by plasma polymerization. The paper shows some novelty and the conclusions are quite well supported by the experimental data.
Some comments and suggestions are listed as follows:
- it could be useful to perform dynamic contact angle measurements, hence evaluating the advancing and receding contact angle values and calculating the hysteresis, which can be directly related to the surface roughness.
- what about the scratch resistance of the deposited coatings?
Author Response
Prof. Kuo Chien-Cheng
National Central University
Chung Da Rd., Chung Li, Taoyuan 32001, Taiwan
Editor-in-Chief
Materials Journal
June 8th, 2022
Dear Reviewer 1,
On behalf of my coauthors, I would like to thank you for the opportunity to revise and resubmit our manuscript ID: materials-1742132, entitled “Superhydrophilic Modification of Polycarbonate Substrate Surface by Organic Plasma Polymerization Film”. We appreciate you and the reviewers for your precious time in reviewing our paper and providing valuable comments. Your valuable and informative comments prompted possible improvements in the current version. We hope that the manuscript will meet your good expectations after careful revisions. If there are any further constructive remarks, the authors would appreciate them.
Below we provide the point-by-point responses. All modifications in the manuscript have been highlighted in the revised file.
- It could be useful to perform dynamic contact angle measurements, hence evaluating the advancing and receding contact angle values and calculating the hysteresis, which can be directly related to the surface roughness.
Response: The model of water contact angle measuring instrument used in our laboratory is not equipped with the image capture function of dynamic contact angle. However, we have use AFM to perform roughness measurements to aid the rationale of this manuscript's discussion of the water contact angle.
- What about the scratch resistance of the deposited coatings?
Response: There is no scratch-resistant analyzer in our laboratory. But according to the literature (H. Zhou et al. Progress in Organic Coatings, 77 (2014), p.1073–p.1078): the Si-O-Si structure formed by cross-linking after APTES cleavage can enhance scratch resistance (3H pencil could not mar the surface). In addition, there are also many literatures proposing the use of APTES to enhance the adhesion between films and substrates (as cited in the reference [2] of this manuscript), it can be seen that this monomer provides strong bonding force after polymerization.
Thank you again for your consideration of our revised manuscript.
Sincerely,
Prof. Kuo Chien-Cheng

Reviewer 2 Report
1- The ratio of Ar/O2 is mentioned to be 2:12.5. What was the exact flow rate of each gas, not their ratio?
2- How was the water contact angle measured? Describe the process more. It is recommended to add some images of the droplets on the substrate showing the water contact angle.
3- Page 6, the XPS spectra and deconvolution of the spectra of the samples are needed.
4- SEM and FTIR analyses may help to improve the quality of the paper.
Author Response
Prof. Kuo Chien-Cheng
National Central University
Chung Da Rd., Chung Li, Taoyuan 32001, Taiwan
Editor-in-Chief
Materials Journal
June 8th, 2022
Dear Reviewer 2,
On behalf of my coauthors, I would like to thank you for the opportunity to revise and resubmit our manuscript ID: materials-1742132, entitled “Superhydrophilic Modification of Polycarbonate Substrate Surface by Organic Plasma Polymerization Film”. We appreciate you and the reviewers for your precious time in reviewing our paper and providing valuable comments. Your valuable and informative comments prompted possible improvements in the current version. We hope that the manuscript will meet your good expectations after careful revisions. If there are any further constructive remarks, the authors would appreciate them.
Below we provide the point-by-point responses. All modifications in the manuscript have been highlighted in the revised file.
- The ratio of Ar/O2 is mentioned to be 2:12.5. What was the exact flow rate of each gas, not their ratio?
Response: During the experiment, the actual flow rate of the gas will vary due to the vacuum degree of the chamber. The flow rate ratio mentioned in the article is the original flow rate of Ar and O2 (Ar: 2 sccm; O2: 12.5 sccm). The narrative in the manuscript will be changed. (Page 2, lines 78-79)
- How was the water contact angle measured? Describe the process more. It is recommended to add some images of the droplets on the substrate showing the water contact angle.
Response: The data of water contact angle are the average values after multiple measurements, so the images of single measurements cannot be attached. The actual measurement of the water droplet images taken will be included in the revised manuscript. (Page 7, Figure 6)
- Page 6, the XPS spectra and deconvolution of the spectra of the samples are needed.
Response: The analysis of XPS was commissioned by other units. This unit only provides the ratio of elements on the surface of our samples. Due to the limited review time, we are apologetic not being able to make and measure samples to provide more information.
- SEM and FTIR analyses may help to improve the quality of the paper.
Response: We have used SEM to measure. Because the electron beam is easy to cause charge accumulation on the polymer, the weak heat resistance of the polymer can easily lead to molecular cleavage, which in turn affects the SEM cavity, so the measured image is blurred. We have also used FTIR for analysis, but the coated film is very thin (only regarded as the surface of the substrate after modification), the qualitative analysis by FTIR cannot be carried out.
Thank you again for your consideration of our revised manuscript.
Sincerely,
Prof. Kuo Chien-Cheng

Reviewer 3 Report
The authors present a solid study on superhydrophobic surfaces created on polycarbonate substrates by means of plasma etching and plasma polymerisation. While the results are interesting, there are some shortcomings in the manuscript that should be addressed prior to publication:
In the materials and methods section, the authors do not describe the pressures during plasma treatments. Although the setup seems to indicate atmospheric pressure, the figure indicates a vacuum vessel. The authors are thus advised to provide working pressures in addition to gas mixtures and electrical parameters.
The authors use the terminology "plasma bombardment", while describing processes as mainly based on chemical reactions in the gas phase. Again, this should be connected to the pressures: if indeed low vacuum pressures are used, the kinetic energies out of the ion sources may be conserved. If, however, mild vacuum or atmospheric pressures are used, the effects should be purely those of chemical reactions by excited species drifting towards the substrates' surfaces. That being said, the authors are asked to provide consistent and accurate descriptions of the processes.
The description of the XPS analysis procedure is lacking details on the measurement parameters (pass energies, etc) and analysis procedures (sources of ionisation cross sections used, etc.).
Moreover, the substrate materials should be properly described (manufacturer or distributer; materials properties or prduct name).
In the presentation of the results, both the description in the text and caption of Figure 4 are wrong, as the displayed contents are not spectra, but rather intensities (normalized to what?) of selected emission lines (which ones? - wavelengths?) dependent on the anode voltage.
Author Response
Prof. Kuo Chien-Cheng
National Central University
Chung Da Rd., Chung Li, Taoyuan 32001, Taiwan
Editor-in-Chief
Materials Journal
June 8th, 2022
Dear Reviewer 3,
On behalf of my coauthors, I would like to thank you for the opportunity to revise and resubmit our manuscript ID: materials-1742132, entitled “Superhydrophilic Modification of Polycarbonate Substrate Surface by Organic Plasma Polymerization Film”. We appreciate you and the reviewers for your precious time in reviewing our paper and providing valuable comments. Your valuable and informative comments prompted possible improvements in the current version. We hope that the manuscript will meet your good expectations after careful revisions. If there are any further constructive remarks, the authors would appreciate them.
Below we provide the point-by-point responses. All modifications in the manuscript have been highlighted in the revised file.
- In the materials and methods section, the authors do not describe the pressures during plasma treatments. Although the setup seems to indicate atmospheric pressure, the figure indicates a vacuum vessel. The authors are thus advised to provide working pressures in addition to gas mixtures and electrical parameters.
Response: The vacuum degree of the cavity of the coating system is 4 × 10-4 Pa. The ion source current and voltage during modification of each sample have been shown in the article. The actual working pressure is about 3 × 10-2 Pa when Ar/O2 is introduced and 2 × 10-1 Pa when N2/APTES is introduced. We will supplement the relevant information in the experimental section of the revised manuscript. (Page 2, line 74; page 2, lines79 – 80; page 2, lines 85 – 86)
- The authors use the terminology "plasma bombardment", while describing processes as mainly based on chemical reactions in the gas phase. Again, this should be connected to the pressures: if indeed low vacuum pressures are used, the kinetic energies out of the ion sources may be conserved. If, however, mild vacuum or atmospheric pressures are used, the effects should be purely those of chemical reactions by excited species drifting towards the substrates' surfaces. That being said, the authors are asked to provide consistent and accurate descriptions of the processes.
Response: The plasma bombardment mentioned in this paper is carried out at low pressure (near vacuum). The vacuum degree of the chamber environment of the coating system is 4 × 10-4 Pa, while the working pressure is about 10-2 to 10-1Pa.
- The description of the XPS analysis procedure is lacking details on the measurement parameters (pass energies, etc) and analysis procedures (sources of ionization cross sections used, etc.).
Response: Since the XPS equipment is measured by other research units, it is more difficult to make samples and measure since the review time is limited. This unit provides only the element ratios on our samples.
- the substrate materials should be properly described (manufacturer or distributer; materials properties or prduct name).
Response: Due to the confidentiality agreements signed with specific manufacturers, detailed information on PC substrates cannot be provided in this study. The plastic substrates used in this study are mainly used for anti-fog applications on the inner side of vehicle lights, so they have undergone rigorous anti-fog and optical tests.
- In the presentation of the results, both the description in the text and caption of Figure 4 are wrong, as the displayed contents are not spectra, but rather intensities (normalized to what?) of selected emission lines (which ones? - wavelengths?) dependent on the anode voltage.
Response: The detected free radical intensities are normalized by Ar plasma at different voltages in the optical emission spectrum, therefore, it can be observed from Figure 4 of this manuscript that Ar is in a straight line, and its normalized intensity is 1. We will revise the description of the re-uploaded manuscript. (Page 5, lines 157-160; page6, description of Figure 4)
Thank you again for your consideration of our revised manuscript.
Sincerely,
Prof. Kuo Chien-Cheng

Round 2
Reviewer 1 Report
The lack of instrumentation in the authors' lab does not allow further improving the manuscript.
Reviewer 2 Report
Most of my comments were not applied.